# Fanconi Anaemia-Like Mph1 Helicase Backs up Rad54 and Rad5 to Circumvent Replication Stress-Driven Chromosome Bridges

**DOI:** 10.3390/genes9110558

**Published:** 2018-11-17

**Authors:** Jonay García-Luis, Félix Machín

**Affiliations:** 1Unidad de Investigación, Hospital Universitario Nuestra Señora de Candelaria, Carretera del Rosario, 145, 38010 Santa Cruz de Tenerife, Spain; 2Instituto de Tecnologías Biomédicas, Universidad de La Laguna, C/Sta María Soledad, 38200 San Cristóbal de La Laguna, Santa Cruz de Tenerife, Spain

**Keywords:** replication stress, methyl methanesulfonate, G2 arrest, anaphase bridges, Mph1 (FANCM), Rad5 (HTLF), Rad54, Sgs1 (BLM), Srs2, fork regression

## Abstract

Homologous recombination (HR) is a preferred mechanism to deal with DNA replication impairments. However, HR synapsis gives rise to joint molecules (JMs) between the nascent sister chromatids, challenging chromosome segregation in anaphase. Joint molecules are resolved by the actions of several structure-selective endonucleases (SSEs), helicases and topoisomerases. Previously, we showed that yeast double mutants for the Mus81-Mms4 and Yen1 SSEs lead to anaphase bridges (ABs) after replication stress. Here, we have studied the role of the Mph1 helicase in preventing these anaphase aberrations. Mph1, the yeast ortholog of Fanconi anaemia protein M (FANCM), is involved in the removal of the D-loop, the first JM to arise in canonical HR. Surprisingly, the absence of Mph1 alone did not increase ABs; rather, it blocked cells in G2. Interestingly, in the search for genetic interactions with functionally related helicases and translocases, we found additive effects on the G2 block and post-G2 aberrations between *mph1*Δ and knockout mutants for Srs2, Rad54 and Rad5. Based on these interactions, we suggest that Mph1 acts coordinately with these helicases in the non-canonical HR-driven fork regression mechanism to bypass stalled replication forks.

## 1. Introduction

Genome stability through multiple cell divisions needs faithful segregation of sister chromatids to the daughter cells. To accomplish this, any linkage that may remain between sisters must have been removed by the time cells enter anaphase. Sister chromatid linkages can be proteinaceous (e.g., cohesion), topological (e.g., catenations) or DNA-DNA mediated (i.e., presence of underreplicated DNA and joint molecules). Joint molecules (JMs) are transient intermediates of the homologous recombination (HR) pathway. In cells undergoing mitotic division, HR works as a reliable error-free DNA repair mechanism against DNA double strand breaks (DSB), stalled replication forks (SRFs), interstrand crosslinks and single stranded DNA (ssDNA) gaps. If sister chromatid linkages are not removed by anaphase, they give rise to anaphase bridges (ABs). In this scenario, the mitotic division may end up in sister chromatid nondisjunction with or without associated DBSs [1]. DSBs at this cell cycle stage are extremely dangerous for the progeny since both ends of the broken DNA lay in different daughter cells and, therefore, there is no way to restore the original DNA molecule by any means.

Joint molecules arise upon synapsis of one DNA strand with the complementary strand of the sister chromatid and can be the consequence of the invasion of one end coming from a DSB, the switch of the leading strand DNA template to bypass a SRF, or else, the switch of the template to fill up a ssDNA gap in one sister chromatid [2,3]. Common to all these variants, the first synaptic event is the formation of a JM termed displacement loop (D-loop). To accomplish synapsis, acceptor DNA must first be single-stranded and coated with the replication protein A (RPA) complex. Then, RPA is displaced by Rad51 to form the ssDNA-Rad51 filament, which starts the search for homology along the donor sister chromatid. Once satisfactory homology is found, the D-loop is shaped by the displacement of the non-complementary donor strand, which then becomes coated by RPA [4,5]. Aside from Rad51, there are other key players during these early HR steps. Thus, important catalysts for DNA synapsis in yeast are Rad52 and other members of its epistasis group such as Rad54. Antagonists of the HR synapsis also exist; for instance, the helicase Srs2 is capable of disassembling the Rad51 filament. Once a stable D-loop has been formed, postsynaptic steps complete HR. In all cases, DNA synthesis will be required. In the synthesis dependent strand annealing (SDSA) pathway, the D-loop remains as the priming substrate for copying the information from the donor sister chromatid and then it is dismantled. Alternatively, the D-loop can be further processed into one or two cross-shaped JM termed Holliday junction (HJ). D-loops and HJs can be resolved by endonucleolytic enzymes or, just in the case of D-loops, by the sole action of DNA helicases [6,7,8]. There is, though, a high degree of specificity for the resolution of each structure. Thus, D-loops are disassembled by the helicase Mph1 (Fanconi anaemia complementation group M, FANCM, in humans), whereas HJs can be resolved by specialised structure selective endonucleases (SSEs), and double HJs (dHJ) by a dissolution pathway which involves the helicase-topoisomerase complex Sgs1-Top3-Rmi1 (STR). This complex is homologous to the human Bloom’s syndrome BLM–TopoIIIa–RMI1–RMI2 complex. In yeast, there are three SSEs able to resolve HJs: the heterodimer complexes Mus81-Mms4 (MUS81-EME1 in humans) and Slx1-Slx4 (SLX1-SLX4/FANCP in humans), and the resolvase Yen1 (GEN1 in humans).

In recent years, a bunch of papers in different organisms have shown that mutants for some of these postsynaptic players form ABs [9,10,11]. Interestingly, proteins such as BLM and FANCM also localised to a new type of DAPI-invisible bridges termed ultrafine anaphase bridges (UFBs) [12,13]. All these bridges happen to be despite HR is tightly coupled to cell cycle checkpoints that prevent anaphase onset upon DNA damage. For instance, the intra-S phase and G2 DNA damage checkpoints can be triggered by conditions that lead to SRFs (e.g., DNA alkylation by methyl methanesulfonate (MMS)). DSBs can also trigger the G2 DNA damage checkpoint. At present, all evidences argue that the RPA-coated ssDNA, which is present in the early processing steps of DSBs by HR and that can also be formed at SRFs, is the universal signal for checkpoint activation and maintenance [14,15]. Nevertheless, since early steps of the HR displace RPA coated ssDNA with Rad51, and then this filament forms JMs upon synapsis, it might be conceptually possible that these checkpoints have been switched off before many JMs are fully resolved. This situation can be exacerbated if cells are challenged to resolve JMs at an increased rate; e.g., during replication stress. Indeed, we previously showed that the double mutant *mms4*Δ *yen1*Δ, deficient in the endonucleolytic resolution of JMs, give rise to DAPI-stained ABs [9]. We further showed that the chromosome XII right arm (cXIIr), which carries the hyperrecombinogenic ribosomal DNA (rDNA) array, was a hot spot for sister chromatid nondisjunction and that this could be rescued by deleting *RAD52*, demonstrating that the source of nondisjunction were HR-driven JMs. Both DAPI-stained ABs and cXIIr nondisjunction occurred apparently blind to cell cycle checkpoints, even under exogenously induced replicative stress. In the present work, we have quantified the formation of DAPI-stained bridges and cXIIr nondisjunction in mutants for several helicases with roles in HR. We have specifically focused on FANCM-like Mph1 and its genetic interactions with the other helicases. We have found that abrogation of Mph1 does not give rise to aberrant anaphase phenotypes but blocks cells at G2. Nevertheless, Mph1 contributes to diminish post-G2 aberrations seen when the activity of Rad5 and Rad54, two helicase members of the SWI/SNF family of DNA translocases, have been impaired.

## 2. Materials and Methods

### 2.1. Yeast Strains, Growth and Experimental Conditions

All yeast strains used in this work are listed in Table 1. Parental YPH499 strain carrying the *cdc15-2* allele, the *tetOs* array at cXIIr-Tel (*tetOs:1061*) and the TetR-YFP fusion has been described elsewhere [16]. Gene deletions and marker swaps were engineered using PCR methods [17].

All strains were grown overnight in air orbital incubators at 25 °C in YEPD (1% w/v Bacto yeast extract, 2% w/v Bacto peptone, 2% w/v glucose) medium. G1-to-telophase experiments were performed as described before [9]. Briefly, asynchronous cultures were first adjusted to an optical density at 600 nm (OD_600_) of 0.5, then synchronised in G1 at 25 °C for 3 h by adding 50 ng mL^−1^ of α-factor (Sigma-Aldrich, cat. no. T6901; all tested strains were *bar1*Δ), and finally released from the G1 arrest at 37 °C for 4 h. For the G1 release, cells were washed twice and resuspended in fresh YEPD containing 0.1 mg mL^−1^ of pronase E (Fluka, cat. no. 81748). Replication stress was induced at the time of the G1 release by adding MMS. MMS was kept in the media until the telophase arrest. In the dose-response experiments, cell cultures were split into ten flasks at the time of the G1 release and nine 1:3 serial dilutions of MMS (Sigma-Aldrich, cat. no. 129925) were used. The MMS final concentrations ranged from 0.1% to 0.000015% v/v (1.2 × 10^−5^ to 1.8 × 10^−9^ M). The tenth culture was left without MMS as a control. To assess experimental reproducibility of the results, the dose-response experiment was at least repeated two more times at three MMS concentrations: 0% (no MMS), borderline concentration “B” and a 1:3 dilution of “B”. Borderline concentration was taken as the highest MMS concentration that allowed >50% of cells to cytologically pass the G2 arrest (mononucleated dumbbell morphology) in our experimental conditions and happened to greatly depend on the mutant background (Table 1, last column).

### 2.2. Single-Cell Analyses by Fluorescence Microscopy

Cell cycle progression, DAPI-stained ABs and cXIIr disjunction were analysed by wide-field fluorescence microscopy [9]. A stack of 20 z-focal plane images (0.3 µm depth) were collected on a Leica DMI6000, using a 63×/1.30 immersion objective and an ultrasensitive DFC 350 digital camera, and processed with the AF6000 software (Leica Microsystems GmbH, Wetzlar, Germany). DAPI (Sigma-Aldrich, cat. no. 32670) was used at 1 µg mL^−1^ final concentration in a 0.25% v/v Triton X-100 (Promega, cat. no. H5141) solution. The staining was performed right before visualisation under the microscope in cell pellets previously frozen at −20 °C for 48 h.

Around 200 cells were quantified per experimental condition and firstly classified according to the following criteria for the cell cycle stage: G1, unbudded cells with the shmoo morphology (indicative of previous response to alpha-factor); G2, mononucleated dumbbells with one body bearing the shmoo morphology; post-G2, shmoo-bearing dumbbells that had either two segregated DAPI masses or one DAPI mass across the bud neck. Post-G2 cells were further analysed for the following anaphase figures: DAPI-bridge, either a single DAPI mass stretched across the bud neck or two DAPI masses when one of them was trailing at the bud neck; cXIIr missegregation, either a single *tetO* focus or two foci found in the same cell body irrespective of the DAPI profile.

### 2.3. Determination of Long-Term MMS Sensitivity

A standard spot assay on YEPD plates was used to assess long-term MMS sensitivity. Briefly, strains were grown overnight in YEPD broth and cell concentration then adjusted to OD_600_ = 10. Next, seven 1:10 serial dilutions were prepared and ~3 µL of all eight samples were spotted on YPD plates supplemented with increasing concentration of MMS. Spotting was carried out with a 48-pin replica plater (Sigma-Aldrich, cat. no. R2383). Growth was determined after incubating at 25 °C for 3 and 7 days. 

### 2.4. Data Representation and Statistics

Error bars in all graphs represent the standard error of the mean (SEM) of three independent biological replicates (N = 3). Comparisons of proportions between different mutants were performed through the unpaired two-tailed Student’s *t* test.

## 3. Results

In previous works, we showed that sister chromatid nondisjunction in a single cell cycle can be precisely detected by fluorescent microscopy in *Saccharomyces cerevisiae*, provided that DAPI-stained ABs are checked together with the segregation status of telomeric regions for the chromosome XII right arm (Tel-cXIIr) [9,16,19]. We had reasoned that Tel-cXIIr segregation status was a good indicator for the overall sister chromatid disjunction since cXIIr appears to be the most segregation-challenging chromosome arm in yeast [19]. This is so, not only because cXIIr is the longest chromosome arm (Figure 1A), but also because it harbours the rDNA array. The rDNA is prone to sister chromatid linkages since it carries natural SRFs, is enriched in spontaneous JMs and is a hot-spot for catenations due to its high levels of transcription [1]. In order to accurately measure Tel-cXIIr segregation, and also stabilise DAPI-stained ABs, we took advantage of the *cdc15-2* thermosensitive allele, which blocks the cells in late mitosis without interfering with chromosome segregation [9,16].

When the G1-to-telophase cell cycle is combined with increasing concentrations of a drug in a dose-response curve, further information can be obtained about the capability of the drug to elicit cell cycle blocks at G1 (unbudded cells) and G2 (mononucleated dumbbells) [20]. By comparing G2 blocks with DAPI-stained ABs and cXIIr nondisjunction in those cells able to pass G2, it is possible to have an overall picture of whether a drug, a mutant, or the combination of drugs and mutants result in anaphase abnormalities unnoticed by the G2 cell cycle checkpoints (Figure 1B). This rationale is the one we previously used to conclude that the double knockout for the SSEs Mms4-Mus81 and Yen1 did not change the cell cycle profile to the replication stress drug MMS, yet it greatly increased the frequency of ABs and cXIIr nondisjunction [9]. This is also the rationale we have followed here to analyse knockout mutants for several DNA helicases involved in HR, or with suspected roles in dealing with replicative stresses (i.e., Mph1, Rad54, Sgs1, Srs2 and Rad5).

### 3.1. Replication Stress Does Not Cause Anaphase Aberrations in Cells Deficient for the Fanconi Anaemia Group M-Like DNA Helicase Mph1

We started this work by comparing the *cdc15-2 mph1*Δ mutant to our reference *cdc15-2* strain. The *mph1*Δ was included in the original screen where SSEs mutants *mms4*Δ and *mms4*Δ *yen1*Δ were studied [9]. Unlike the SSEs mutants, the G2 arrest profile in *mph1*Δ was different to the reference strain (Figure 2A, upper charts); whereas MMS addition gave a single peak of G2 arrest at 0.01% v/v in the reference *cdc15-2* strain and the SSEs mutants, the G2 arrest was still prevalent in *cdc15-2 mph1*Δ at MMS concentrations 3-fold lower (Figure 2B). This fact made it impossible to compare anaphase figures at 0.004% v/v, as we had done before for the SSEs mutants. Such “borderline” concentration of MMS had been chosen because it was the highest that allowed most cells to enter anaphase from a cytological point of view; that is, equal or greater than 50% of cells were either binucleated dumbbells or with a stretched DAPI bridge across the bud neck. However, the dose-response MMS curve let us set the borderline concentration for *cdc15-2 mph1*Δ at 0.001% (Table 1), and still compared the anaphase figures between borderline concentrations. Strikingly, neither DAPI-stained bridges (<10% of post-G2 cells) nor cXIIr missegregation (<20%) was higher than the corresponding control without MMS (Figure 2A, lower charts; and 2C). This anaphase profile was not different to that of the reference *cdc15-2* strain, yet significantly different to the SSEs mutants; particularly the *mms4*Δ *yen1*Δ double mutant at the borderline concentration (~80% of post-G2 cells missegregated cXIIr and ~20% had DAPI bridges).

In addition to the G1-to-telophase cell cycle profile, we compared the long-term sensitivity to MMS of these mutants through spot assays. We observed that *cdc15-2 mph1*Δ was only slightly more sensitive than the reference *cdc15-2* strain (Figure 2D). By contrast, *cdc15-2 mms4*Δ *yen1*Δ was rather hypersensitive and *cdc15-2 mms4*Δ was still more sensitive than *cdc15-2 mph1*Δ. The only mild increase in MMS sensitivity of *mph1*Δ is likely a consequence of the compensatory effect of bypassing SRFs by the mutagenic translesion synthesis (TLS) pathway, as it has been shown before in this mutant [21,22].

### 3.2. The G2 and Post-G2 Profiles of mph1Δ are Hypostatic to rad51Δ but Additive to rad54Δ

The fact that *cdc15-2 mph1*Δ brought about a broader G2 peak suggests that, under replicative stress, a stronger G2 checkpoint is triggered in this mutant; i.e., more RPA-ssDNA could be present as a result of replication fork stalling by MMS. The steady-state levels of RPA-ssDNA are negatively controlled by the efficiency of the transition between the RPA-ssDNA and the Rad51-DNA filaments [23]. We therefore compared the *cdc15-2 mph1*Δ profiles to those seen in the presynaptic *cdc15-2 rad51*Δ mutant. We found that the G2 profile in the *cdc15-2 rad51*Δ MMS dose-response curve was even broader than that of the *cdc15-2 mph1*Δ strain (Figure 3A, leftmost upper chart; Table 1). In addition, *rad51*Δ, but not *mph1*Δ, yielded DAPI bridges in up to 20% of post-G2 cells at the borderline MMS concentration (Figure 3A, leftmost lower chart, and Figure 3B). Of note, the morphology of the DAPI-stained bridge in *rad51*Δ tended to show two highly interconnected nuclear masses across the bud neck (Figure 3C), as opposed to the elongated DAPI bridge seen in the *cdc15-2 mms4*Δ *yen1*Δ mutant. This happened in at least 75% of the observed *cdc15-2 rad51*Δ DAPI-bridges. This bow-tie morphology suggests a highly tight anaphase DAPI-stained bridge, as the one observed in *top2-4 cdc15-2* [24]. Furthermore, cXIIr missegregation alone (i.e., without a concomitant visible DAPI bridge) was rare in this mutant (<10% of post-G2 cells). Similar cell biology phenotypes were seen in *cdc15-2 rad52*Δ (data not shown). Finally, the long-term MMS sensitivity showed remarkable differences between *cdc15-2 mph1*Δ and *cdc15-2 rad51*Δ; with the latter being strongly hypersensitive to MMS (Figure 3D).

We next included the *cdc15-2 rad54*Δ mutant in this analysis. Rad54 is an ATP DNA translocase and an important catalyst in the disassembly of the Rad51 filament once the DNA synapsis with the donor template takes places, hence facilitating the creation of a functional D-loop [25]. Dose-response MMS profiles for *cdc15-2 rad51*Δ and *cdc15-2 rad54*Δ were entirely equivalent (Figure 3A,B). Just regarding the G2 profile, this result suggests that during the replication stress the G2 checkpoint response is highly active even after disassembling the RPA-ssDNA filament in favour of the Rad51-ssDNA filament.

Finally, we compared the double *mph1*Δ *rad51*Δ and *mph1*Δ *rad54*Δ mutants with the corresponding single mutants. The segregation defects (bow-tie DAPI bridges), G2 block profiles and long-term hypersensitivity to MMS of *cdc15-2 mph1*Δ *rad51*Δ were equivalent to those of *cdc15-2 rad51*Δ (Figure 3A,B,D). This equivalence establishes an epistatic relationship that positions Mph1 downstream Rad51 within the same pathway, as well as suggesting that Mph1 would have a compensatory activity that makes the *mph1*Δ mutant milder than presynaptic HR mutants to replication stress. The long-term hypersensitivity and G2 profiles were also similar between *cdc15-2 mph1*Δ *rad54*Δ and *cdc15-2 rad54*Δ, but an additive post-G2 effect was observed at 0.0004% MMS (Figure 3B). This non-epistatic relationship suggests that Mph1 backs up Rad54 in dealing with replication stress intermediates that may compromise mitotic division.

### 3.3. Abrogation of the Functionally Related Helicase Srs2 Enhances the mph1Δ G2 Block without Causing More Aberrant Post-G2 Figures

Mph1 is a 3′-to-5′ helicase that belongs to the SF2 superfamily. Srs2 is another SF2 3′-to-5′ helicase which has been clearly implicated in HR and could theoretically complement Mph1. Indeed, Srs2 is able to biochemically dismantle D-loops and has a strong negative genetic interaction with Mph1 [26,27]. Consequently, we decided to study the G2 and post-G2 profiles of *cdc15-2* mutant derivatives for *SRS2*, compare it to *cdc15-2 mph1*Δ, and derive their epistatic relationships through the analysis of the *cdc15-2 mph1*Δ *srs2*Δ double mutant (Figure 4). We found that the cell cycle profile to increasing MMS concentrations of *cdc15-2 srs2*Δ was reminiscent of *cdc15-2*; i.e., a G2 peak at 0.01% MMS, although a slight increase in G2 blocks at lower MMS concentrations were also observed (Figure 4A). However, The G2 profile of the *mph1*Δ *srs2*Δ double mutant was synergistic, with a borderline at 0.0004% v/v, 3-fold lower than *mph1*Δ. By contrast, the post-G2 segregation figures of *mph1*Δ *srs2*Δ and *srs2*Δ were equivalent (Figure 4A,B), despite *srs2*Δ having an increase in cXIIr missegregation relative to *mph1*Δ (~30% vs. ~15%). The long-term hypersensitivity to MMS showed a clear and synergistic negative interaction between *mph1*Δ and *srs2*Δ (Figure 4C), confirming previous reports [22,27,28].

### 3.4. The G2 and Post-G2 Profiles of mph1Δ Are Neutral to Those Observed in sgs1Δ

A third SF2 3′-to-5′ helicase implicated in HR is Sgs1, the yeast ortholog of human BLM. Sgs1 works within the Sgs1-Top3-Rmi1 (STR) helicase-topoisomerase complex as a dHJ dissolvasome. As such, it has been implicated in the removal of late JMs, after the maturation of the D-loop into HJs [29]. However, BLM has also been shown to dissolve D-loops in a mechanism that might complement FANCM/Mph1 activity [30,31]. We found that the cell cycle profile to increasing MMS concentrations of *cdc15-2 sgs1*Δ was like *cdc15-2*; i.e., a single G2 peak at 0.01% MMS (Figure 5A). The G2 profile of the *mph1*Δ *sgs1*Δ double mutant was more similar to *mph1*Δ than was to *sgs1*Δ, whereas the post-G2 segregation figure was more similar to *sgs1*Δ at the corresponding MMS borderline concentrations (Figure 5A,B). Noteworthy, *sgs1*Δ showed a clear dose-response profile of anaphase abnormalities, rather reminiscent of that observed for the SSE *mms4*Δ *yen1*Δ double mutant, but with just half of their values. These cell cycle profiles likely position Sgs1 at a stage wherein timely removal of late JMs is a must to prevent mitotic catastrophe. In addition, they recapitulate our previous findings with SSE mutants, indicating that JMs that serve as substrates for both Sgs1 and SSEs are not sense by the G2 checkpoints [9]. The facts that (i) there is no extensive G2 peak in *sgs1*Δ and (ii) the G2 peak profile of *mph1*Δ *sgs1*Δ is epistatic to *mph1*Δ, situate Mph1 upstream Sgs1 in a linear pathway devoted to bypassing SRFs. In addition, the facts that (i) there is post-G2 segregation defects in *sgs1*Δ, not seen in *mph1*Δ, and (ii) the post-G2 profile of *mph1*Δ *sgs1*Δ is epistatic to *sgs1*Δ, suggests that Mph1 has no role in removing late JMs, contrary to our initial expectations. Finally, although our epistatic analysis for short-term phenotypes strongly suggests that Sgs1 and Mph1 do not co-work on the same substrates, we observed an additive effect on long-term MMS sensitivity (Figure 5C). These results suggest that additive fitness defects in the double mutant might arise from successive cell divisions under suboptimal conditions to confront replication stress; e.g., the consequences of the higher cXIIr missegregation in *sgs1*Δ could be ameliorated by Mph1 in the following cell cycles, etc.

### 3.5. The Rad5 Helicase Prevents Aberrant post-G2 Figures upon Low Replication Stress and Mph1 Backs Up These Rad5 Safeguard Functions

The stronger MMS-dependent G2 arrest in *mph1*Δ, the lack of cumulative aberrant post-G2 figures in double mutants with 3′-5′ helicases, together with the synergistic post-G2 interaction with *rad54*Δ, led us to hypothesise that Mph1’s main role may lay at the beginning of the tolerance pathways that deal with replicative stress. In fact, Mph1 and its orthologs have been implicated in the regression of the SRF [32,33], as it has been also shown for the human Rad54 homolog [34]. This mechanism is used to bypass the lesion by allowing DNA synthesis using the nascent lagging strand as a template for the leading strand, once this strand is left behind by the blockage. The regressed fork can then be either deregressed or process into other structures that are metabolised by the HR pathway [5,35,36]. Rad5 and its human tumour-suppressor ortholog HLTF, have been implicated in fork regression as well [35,37,38]. We therefore studied the cell cycle MMS profiles of *cdc15-2 rad5*Δ and *cdc15-2 mph1*Δ *rad5*Δ. Outstandingly, *cdc15-2 rad5*Δ behaved like *cdc15-2 rad51*Δ and *cdc15-2 rad54*Δ, yet the post-G2 aberrant phenotypes were even stronger (Figure 6A,B). Likewise, most post-G2 abnormalities comprised bow-tie nuclei. Combination of *rad5*Δ and *mph1*Δ caused an additive effect; i.e., more post-G2 figures at the MMS borderline concentration. Long-term MMS sensitivity of *rad5*Δ and *mph1*Δ *rad5*Δ phenocopied those of *rad51*Δ and *rad54*Δ combinations, including their epistatic relationships (Figure 6C). We conclude that Rad5 plays an important role for a successful tolerance to replication stress, with Mph1 backing up Rad5 for a safe G2/M transition.

## 4. Discussion

In this work, we have characterised the cell cycle response and the post-G2 (anaphase) phenotypes that replication stress by MMS cause in a set of mutants for the HR pathway. Importantly, we have tested not a single MMS concentration but multiple scenarios that cover up to 4 orders of magnitude for MMS concentrations. This approach has let us overcome the caveat of having different G2 checkpoint sensitivities to MMS among the studied mutants. We have thus defined G2/post-G2 borderline concentrations for each mutant combination and further characterised the putative post-G2 aberrations our strains are designed for; i.e., gross chromatin anaphase bridges (DAPI bridges) and nondisjunction of the segregation-challenging chromosome XII right arm (cXIIr). The borderline parameter is almost equivalent to the EC_50_ used in pharmacology and is likely the most valid approach to compare the post-G2 figures among mutants with dissimilar sensitivity to trigger and/or sustain the G2 checkpoints. In addition, the broad range of MMS concentrations allows to compare the effect of low and high replication stress since differences between these regimes, in terms of genetic pathways required for damage tolerance and survival, have been raised in recent years [39,40].

Although we have mainly focused on strains deficient for the Mph1 helicase (ortholog of FANCM in humans), we have also characterised several single knockout mutants for HR players during the analysis of Mph1’s genetic interactions. Just by comparing their MMS dose-response profiles, we can group our single mutants in the following categories (Figure 7A). First, mutants that behave like that the reference “wild type” strain; i.e., only being able to trigger a sustained G2 block at 0.01% v/v MMS, with a corresponding borderline at 0.004% v/v and with a low baseline of DAPI bridges (<10%) and cXIIr missegregation (<20%): *srs2*Δ. Second, mutants that differ from the former only in the abnormal post-G2 figures: *sgs1*Δ and *mms4*Δ. Third, mutants that yielded a moderate G2 block profile, with G2/post-G2 borderlines around 0.001% v/v, and no abnormal post-G2 figures: *mph1*Δ. Fourth, mutants that yielded a much stronger G2 block profile, with G2/post-G2 borderlines at or below 0.0004% v/v, and with significant bow-tie DAPI bridges at the borderline: *rad5*Δ, *rad51*Δ and *rad54*Δ. The bow-tie DAPI bridge is a phenotype that seems to reside in the transition between G2 and anaphase. Bow-tie nuclei get enriched after DSB generation in incomplete mutants for the G2 DNA damage checkpoint (e.g., *rad53*Δ and *chk1*Δ), and are reported to be able to dynamically go back to a G2 state (mononucleated dumbbell cell) [41]. However, similar bow-tie phenotypes are seen in *cdc15-2 top2-4*, known to enter anaphase with catenations that tightly linked sister chromatids [24]. It is worth noting that this particular phenotype was present in those mutants that show the highest sensitivity to MMS, both in the G2 arrest profiles and in the long-term survival. Furthermore, it was dependent on the MMS concentration, as there was a shift between the bow-tie nucleus and a proper G2 block as MMS concentration went higher. The best explanation for this direct relationship is that the increasing amount of MMS-driven sister chromatid linkages would result in tighter anaphases that cytologically appear as bow-tie.

Regarding the cell biology phenotypes of the *mph1*Δ double mutants, we must highlight that most genetic interactions were epistatic. With respect to its contribution in the activation of and later recovery from the G2 checkpoints, these interactions posit Mph1 in linear pathways wherein Mph1 lays downstream of Rad51, Rad54 and Rad5, while laying upstream of Sgs1. However, post-G2 segregation figures surprisingly position Mph1 out of the Sgs1 pathway. To fit the observed phenotypes and Mph1’s genetic interactions in the context of the current models of DNA damage tolerance (DDT) to replication stress is challenging since the molecular mechanisms underlying DDT are far from being understood. Nevertheless, we can speculate about the roles of the different players based on the results we present here and previous literature. The novel key point is that the most evident Mph1 function takes place at early stages (presynaptic) of the HR-driven error-free DDT subpathways. The non-epistatic interactions with Rad5 and Rad54, additive in the persistence of the G2 block and synergistic in the post-G2 profiles (Figure 2, Figure 3, Figure 6 and Appendix A and Table 1), lead us to place Mph1 in the of RF regression subpathway (Figure 7B). This subpathway is proposed to operate when the leading strand gets blocked by a barrier (e.g., an MMS-derived DNA adduct) [35,36]. Lagging strand would continue to grow in length and, as a result, ssDNA would arise in front the blocked leading strand. This ssDNA, immediately coated by the RPA complex, would trigger the G2 checkpoint. Next, the RPA-ssDNA would turn into the ssDNA-Rad51 filament and this would prepare the SRF for fork regression and annealing of the nascent lagging and leading strands. Fork regression would be possible by the helicase and/or translocase activities of Mph1, Rad5 or Rad54 [33,37]. Even though fork regression activity has not been demonstrated for yeast Rad54, its human homolog RAD54 owns such activity in partnership with hRAD51 [34], supporting the inclusion of Rad54 in this model. The synergistic post-G2 profiles of *mph1*Δ with *rad5*Δ and *rad54*Δ, would fit in this model provided that Mph1 backs up either helicase in the fork regression activity. In addition, Mph1 could collaborate with Srs2 in removing Rad51 to favour maturation of the regressed RF for DNA synthesis. This would explain the negative synergism of *mph1*Δ *srs2*Δ, with G2 block and long-term fitness profiles seemingly similar to the *rad* mutants. It is also plausible that these helicases/translocases deregress the regressed RF [38], bypassing the MMS abduct, and turning off the G2 checkpoint. Lastly but not least, our results pinpoint a backup role of Mph1, especially in relation to Rad5 and Rad54. Thus, the *mph1*Δ mutant is intermediate in the G2 block profile and only slightly more long-term hypersensitive than the wild type (WT). We propose that both Rad54 and Rad5 lead the HR-driven DDT to MMS. To fit in this model the TLS-dependent hypermutagenic phenotypes of *mph1*Δ [21], we also propose that RF regression (or any other HR-driven subpathway) is attempted before committing cells to bypass the SRF through TLS. In the absence of Mph1, some SRFs cannot be regressed on schedule, or lead to problematic intermediates, and are diverted to be bypassed through TLS. However, in the absence of Rad5, and other presynaptic HR players, SRFs cannot be processed correctly, not even diverted towards TLS, maybe because they end up collapsing and breaking apart. Of course, the overall picture is probably more complicated since the model will need to accommodate the contribution of these players in the fate of the ssDNA gaps left behind upon blockage of the lagging strand as well as alternative mechanisms to bypass SRFs, including those that envision D-loop structures ahead of the SRF [36].

Finally, it is intriguing that only *mms4*Δ, *mms4*Δ *yen1*Δ and *sgs1*Δ gave rise to enhanced cXIIr missegregation profiles; i.e., an increase in the ratio between cXIIr missegregation and DAPI bridges (~4:1 in these mutants versus <2:1 in the other mutants and WT). This enhancement suggests that replication stress is bypassed at the rDNA through more canonical HR mechanisms, such as template switching that results in dHJs. In this context, it is worth mentioning that RF regression might be repressed within this repetitive locus [42,43].

In conclusion, in this work we demonstrate that the helicase Mph1 (FANCM) contributes to the short-term tolerance to replication stress. We conclude that Mph1 deficiency leads to an extended G2 block upon replication stress and more aberrant post-G2 segregation figures when combined with deficiencies for related helicases/translocases such as Rad5 and Rad54. We believe that the plethora of short-term interactions we show here provide interesting insights for future combined interventions in the new era of personalised anti-cancer chemotherapy.

## Figures and Tables

**Figure 1 genes-09-00558-f001:**
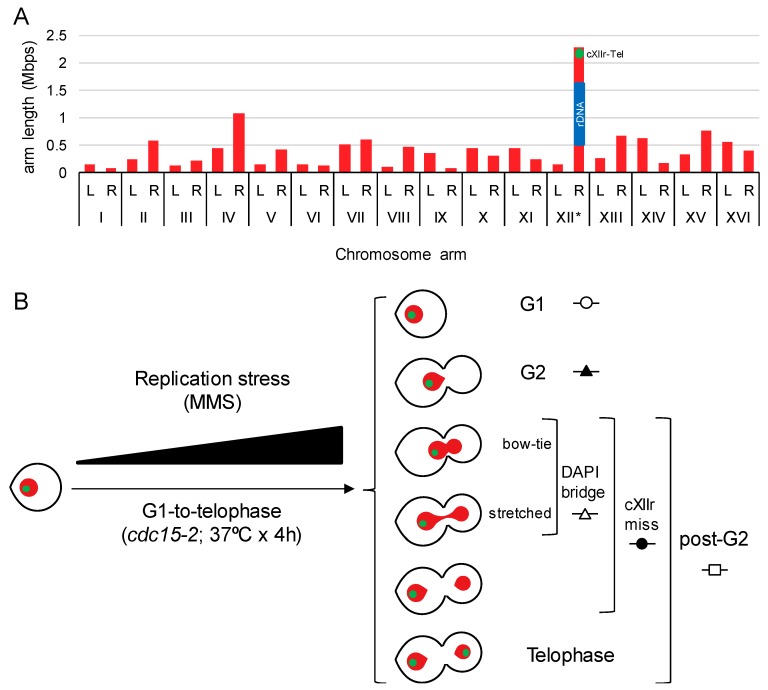
Yeast chromosome arm lengths and cell morphologies obtained after a G1-to-telophase synchronous cell cycle. (**A**) Chart of S288C chromosome arm lengths. Length of chromosome XII right arm (cXIIr) in an estimate considering 150 copies of the 9.1 Kb ribosomal DNA (rDNA) unit. L, left arm; R, right arm; blue rectangle, rDNA array; green dot, position of the engineered *tetOs* array present in all strains of this study. (**B**) Schematic of the G1-to-telophase experiments, undertaken in this study under increasing doses of replication stress, and the main end-point cell figures considered for analysis. Red, main nuclear mass stained by DAPI, green spot, cXIIr subtelometic region (Tel-cXIIr) as seen by the *tetOs*/tetR-YFP reporter.

**Figure 2 genes-09-00558-f002:**
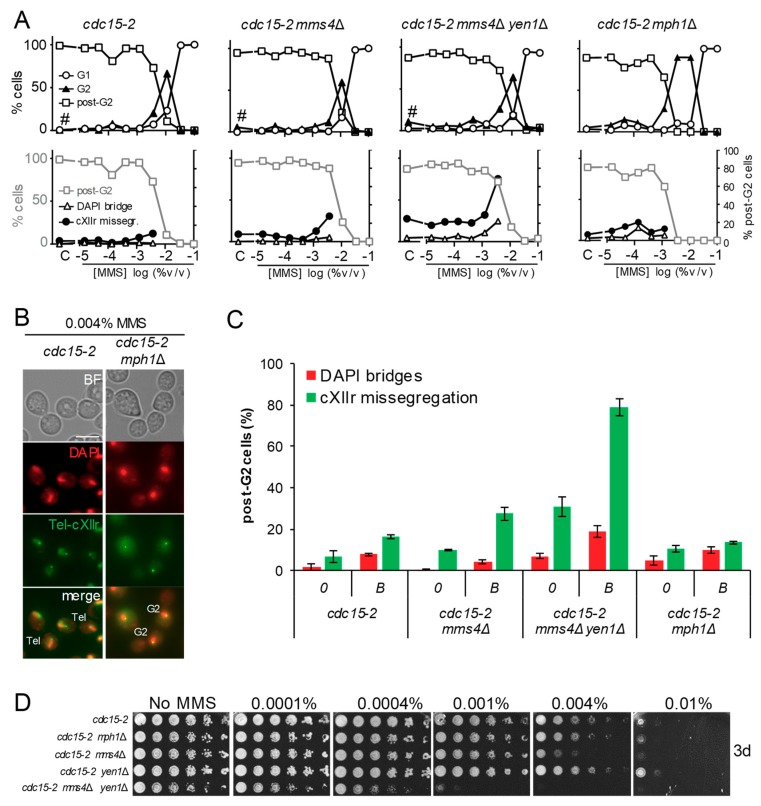
Deletion of *MPH1* enhanced an MMS-induced G2 block without causing anaphase aberrations. (**A**) MMS dose-response curves of the reference strain FM588 (*cdc15-2 Tel-cXIIr:tetO tetR-YFP*), knocked out single mutant derivatives for *MMS4* and *MPH1*, and the double mutant *mms4*Δ *yen1*Δ. The responses were (i) end-point cell cycle profiles after a G1-to-telophase release for 4 h (upper charts) and (ii) aberrant figures in those cells able to pass G2 by 4 h (lower charts; secondary y-axis). In the lower charts, percentage of post-G2 cells is shown again in grey as a reference (primary y-axis). The MMS dose range was from 0.1% to 0.000015% v/v in 1:3 serial dilutions plus a control (“C”) without MMS (x-axis). The hash (#) indicates charts already included in the supplemental material of [9], shown here as a reference for the other related charts. (**B**) Micrographs of representative cells at 0.004% v/v MMS. Tel, telophase cells (binucleated dumbbells without DAPI-bridges and with segregated Tel-cXIIr); G2, cells blocked at G2 (mononucleated dumbbells). Scale bars represents 5 µm. (**C**) DAPI-bridges and cXIIr missegregation of three independent experiments without (“0”) or with MMS (mean ± SEM). Only a single MMS concentration was considered for the analysis; i.e., the highest that allowed >50% of cells to pass G2 (borderline or “*B*” concentration): 0.004% v/v for the reference strain, *mms4*Δ, and *mms4*Δ *yen1*Δ; 0.001% v/v for *mph1*Δ. (**D**) Long-term survival (spotting assay) for the referred mutants under increasing doses of MMS.

**Figure 3 genes-09-00558-f003:**
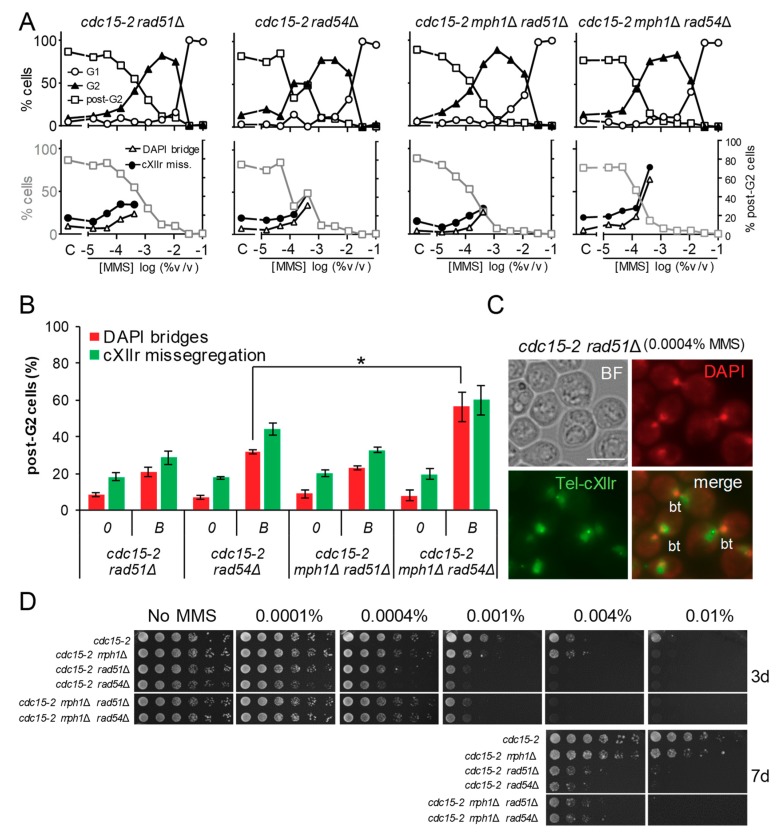
Deletion of *MPH1* was hypostatic to *rad51*Δ but additive to *rad54*Δ regarding post-G2 aberrant figures. (**A**) MMS dose-response curves of FM588 single knocked out mutant derivatives for *RAD51* and *RAD54*, as well as the double mutants *mph1*Δ *rad51*Δ and *mph1*Δ *rad54*Δ. (**B**) DAPI-bridges and cXIIr missegregation of three independent experiments without (“0”) or with borderline (“*B*”) MMS (mean ± SEM). *B* was 0.0004% v/v in all cases. The asterisk (*) highlights the comparison of percentages of DAPI bridges between *rad54*Δ and *mph1*Δ *rad54*Δ (*p* < 0.05, Student’s *t* test). (**C**) Micrographs of representative *rad51*Δ cells at 0.0004% v/v MMS. “bt” refers to “bow-tie” phenotype: dumbbell cell with a short DAPI bridge across the budneck. Scale bars represents 5 µm. (**D**) Long-term survival (spotting assay) for the referred mutants under increasing doses of MMS.

**Figure 4 genes-09-00558-f004:**
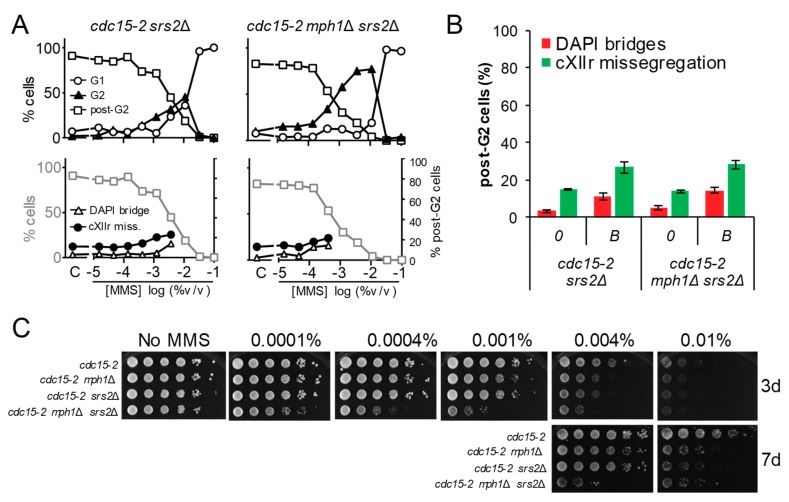
The elimination of the 3′ → 5′ helicase Srs2 exacerbates the MMS-driven G2 block of *mph1*Δ. (**A**) MMS dose-response curves of the FM588 *srs2*Δ and *srs2*Δ *mph1*Δ mutant derivatives. (**B**) DAPI-bridges and cXIIr missegregation of three independent experiments without (“0”) or with borderline (“*B*”) MMS (mean ± SEM). *B* was 0.004% v/v for *srs2*Δ and 0.0004% v/v for *srs2*Δ *mph1*Δ. (**C**) Long-term survival (spotting assay) for the referred mutants under increasing doses of MMS. Note the negative synergism between *srs2*Δ and *mph1*Δ with regards to the short-term G2 block and long-term fitness.

**Figure 5 genes-09-00558-f005:**
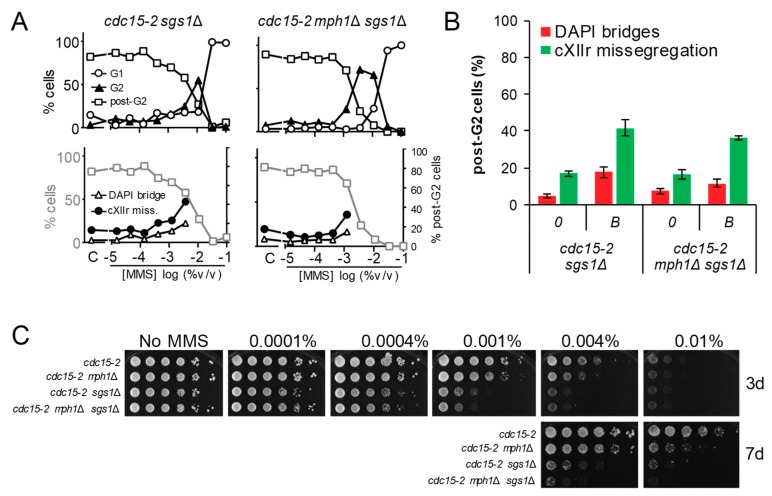
The combination of *mph1*Δ and the knockout for the 3′ → 5′ helicase Sgs1 renders neutral genetic interactions with respect to the G2 block and post-G2 figures. (**A**) MMS dose-response curves of the FM588 *sgs1*Δ and *sgs1*Δ *mph1*Δ mutant derivatives. (**B**) DAPI-bridges and cXIIr missegregation of three independent experiments without (“0”) or with borderline (“*B*”) MMS (mean ± SEM). *B* was 0.004% v/v for *sgs1*Δ and 0.001% v/v for *sgs1*Δ *mph1*Δ. (**C**) Long-term survival (spotting assay) for the referred mutants under increasing doses of MMS. Note that, relative to *sgs1*Δ, *mph1*Δ was (i) epistatic for the G2 block, (ii) hypostatic for cXIIr missegregation and (iii) synthetic (negative synergism) for long-term fitness at high, but not low, MMS concentrations.

**Figure 6 genes-09-00558-f006:**
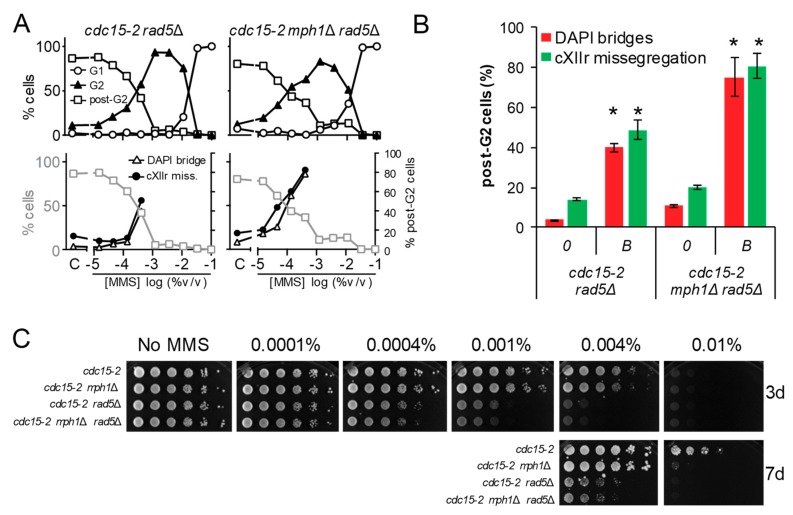
Deletion of *MPH1* synergistically increases the post-G2 aberrant figures observed in *rad5*Δ. (**A**) MMS dose-response curves of the FM588 *rad5*Δ and *rad5*Δ *mph1*Δ mutant derivatives. (**B**) DAPI-bridges and cXIIr missegregation of three independent experiments without (“0”) or with borderline (“*B*”) MMS (mean ± SEM). *B* was 0.0004% v/v for both strains. Asterisks (*) highlight the comparisons of percentages of DAPI bridges and cXIIr missegregation between *rad5*Δ and *mph1*Δ *rad5*Δ (*p* < 0.05, Student’s *t* test). (**C**) Long-term survival (spotting assay) for the referred mutants under increasing doses of MMS.

**Figure 7 genes-09-00558-f007:**
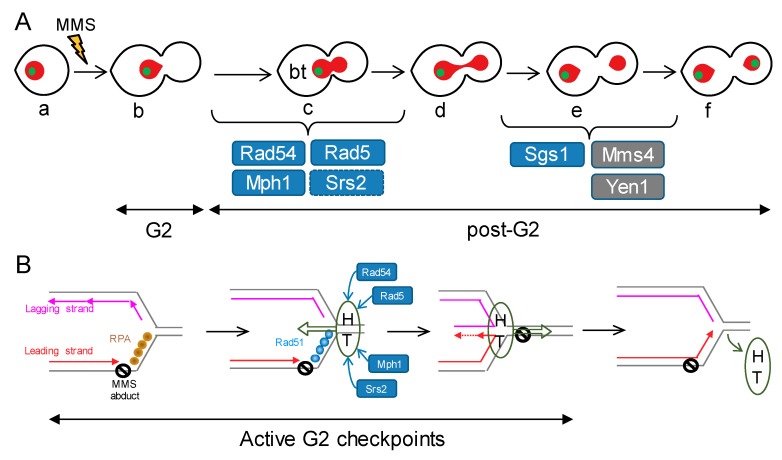
Models of how Mph1 might cooperate with other HR helicases/translocases during replication stress in order to avoid later chromosome segregation problems. (**A**) Putative steps during the mitotic progression in which the studied factors may exert a safeguard role after replication stress. MMS would cause SRFs and activation of G2 checkpoints, rendering the prototypical mononucleated dumbbell cell (b). HR-dependent bypass/repair of the SRFs would turn off the G2 checkpoint, allowing cells to enter anaphase and attempt sister chromatid segregation (c–f). Removal of HR-dependent JMs is a must to achieve proper segregation (f). Rad54 and Rad5 are thus essential for bypassing/repairing the SRFs, switching off the G2 checkpoint, and entering an apparently normal anaphase. Mph1 safeguard function also resides at the G2-anaphase transition, yet to a lesser extent. Srs2 backs up Mph1 here, whereas Mph1 backs up Rad5 and Rad54. Pathways that deal with HJs (one kind of JMs), namely Mms4-Mus81, Yen1 and Sgs1, appear especially important for cXIIr segregation (e–f transition). Blue boxes depict helicases/translocases, whereas grey boxes depict SSEs. (**B**) Model of co-operation among the studied factors in the context of the SRF bypass through the fork regression mechanism. An MMS adduct on the parental template for the leading strand would cause loss of replication synchrony at the affected RF. The exposed ssDNA ahead of the leading strand block would be coated by RPA, triggering a G2 block and favouring repair by HR. Rad51 would replace RPA shortly afterwards. Fork regression could be executed through the co-ordinated actions of HR factors together with helicases and translocases. For instance, a helicase-translocase complex (HT) could translocate de RF backwards (Rad5, Rad54 and Mph1). Rad51 would promote rapid annealing of the nascent strands, a likely step needed to nucleate and sustain the fork regression reaction. Srs2 and Mph1 would remove Rad51 while fork regression is in progress, favouring the formation of the cross-shaped intermediate that serves as template for extension of the leading strand. Once this step has been accomplished, the same (or part of the) helicase-translocase complex would shift direction, translocating the regressed RF into a proper RF, thus bypassing the lesion on the parental strand and turning off the G2 checkpoint.

**Table 1 genes-09-00558-t001:** Strains used in this study and their corresponding methyl methanesulfonate (MMS) borderline concentrations.

Strain	Genotype	Origin	MMS Borderline ^c^
AS499 (YPH499)	*MATa ura3-52 lys2-801 ade2-101 trp1-Δ63 his3-Δ200 leu2-Δ1 bar1-Δ*	[18] ^a^	n.d.
FM588	AS499; *ade2-101:TetR-YFP:ADE2; cXIIr(1061Kb):tetOs:HIS3*; *cdc15-2:9myc:Hph*	[16] ^b^	0.004%
FM992	FM588; *Δmms4::KanMX*	[9]	0.004%
FM1185	FM588; *Δmms4::KanMX*; *Δyen1::BleMX*	[9]	0.004%
FM826	FM588; *Δmph1::KanMX*	This work	0.001%
FM987	FM588; *∆rad51::kanMX4*	This work	0.0004%
FM982	FM588; *Δmph1::NatMX*; *∆rad51::KanMX4*	This work	0.0004%
FM833	FM588; *Δrad54::KanMX*	This work	0.0004%
FM1044	FM588; *Δmph1::NatMX*; *∆rad54::KanMX4*	This work	0.0004%
FM690	FM588; *Δsgs1::KanMX*	This work	0.004%
FM999	FM588; *Δmph1::NatMX*; *∆sgs1::KanMX4*	This work	0.001%
FM837	FM588; *Δsrs2::KanMX*	This work	0.004%
FM1026	FM588; *Δmph1::NatMX*; *∆srs2::KanMX4*	This work	0.0004%
FM934	FM588; *Δrad5::KanMX*	This work	0.0004%
FM1028	FM588; *Δmph1::NatMX*; *∆rad5::KanMX4*	This work	0.0004%

^a^ Parental strain; it is a *bar1* derivative of the S288C congenic strain YPH499 [18]. ^b^ Reference strain [16]. ^c^ MMS concentrations (% v/v) used in the post-G2 comparative analyses. In general, borderline concentration was defined as the maximum concentration assayed which still yielded more than 50% post-G2 cells in at least 2 out of 3 independent repeats of the G1-to-telophase experiment. In practice, the borderline for some double mutants was raised in order to perform more equal comparisons (e.g., *Δmph1 ∆rad54*; *Δmph1 ∆rad5*). In such cases, post-G2 cells were ~25%.

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
