# Peer review of "Fanconi Anaemia-Like Mph1 Helicase Backs up Rad54 and Rad5 to Circumvent Replication Stress-Driven Chromosome Bridges"

_genes, 2018, doi:10.3390/genes9110558_

Round 1

Reviewer 1 Report

Garcia-Luis J et al have conducted a systematic analysis of cell cycle response and the post –G2 (anaphase) phenotype, including gross chromatin anaphase bridge (DAPI bridge) and cXIIr mis-segregation, in response to replication stress induced by MMS in single and compound mutants of MPH1 and other nucleic acid motor proteins and homologous recombination factors. The authors have found that mph1 does not cause anaphase aberrations in response to replication stress but causes G2 arrest. They have further determined the genetic interactions of Mph1 and other related DNA helicases and translocases and shown that co-depletion of Mph1 with Rad54, Srs2 and Rad5 has an additive effects on G2 phase arrest and post-G2 aberrations. In general, the data support the conclusions and the study provides evidence that nucleic acid motor proteins including Mph1, Rad5 and Rad54 function in preventing the accumulation of chromosome aberrations induced by replication stress. Please address the following points: 1) Fig. 2-6A, the authors should label these open circle, open squares, etc., with a definition in each case to help readers understand what is shown in the figure. 2) Label the first panel in Fig. 2 “A”. 3) What is the rationale to compare the cdc15-2, cdc15-2 SSE mutants with cdc15-2 mph1 delta at different MMS concentrations? The authors should show cdc15-2 mph1 delta at 0.004% MMS concentration in one of supplemental figure.

Author Response

(authors’ replies in bold)

Garcia-Luis J et al have conducted a systematic analysis of cell cycle response and the post –G2 (anaphase) phenotype, including gross chromatin anaphase bridge (DAPI bridge) and cXIIr mis-segregation, in response to replication stress induced by MMS in single and compound mutants of MPH1 and other nucleic acid motor proteins and homologous recombination factors. The authors have found that mph1D does not cause anaphase aberrations in response to replication stress but causes G2 arrest. They have further determined the genetic interactions of Mph1 and other related DNA helicases and translocases and shown that co-depletion of Mph1 with Rad54, Srs2 and Rad5 has an additive effects on G2 phase arrest and post-G2 aberrations. In general, the data support the conclusions and the study provides evidence that nucleic acid motor proteins including Mph1, Rad5 and Rad54 function in preventing the accumulation of chromosome aberrations induced by replication stress. Please address the following points:

1) Fig. 2-6A, the authors should label these open circle, open squares, etc., with a definition in each case to help readers understand what is shown in the figure.

The definition of these labels is in both Figure 1B and Figure 2 legend already. However, we have now included the requested visual legends in all these Figures.  

2) Label the first panel in Fig. 2 “A”.

Corrected.

3) What is the rationale to compare the cdc15-2, cdc15-2 SSE mutants with cdc15-2 mph1 delta at different MMS concentrations? The authors should show cdc15-2 mph1 delta at 0.004% MMS concentration in one of supplemental figure.

The rationale was to compare concentrations at the borderline (EC50) between G2 block and post-G2. The point is that mph1Δ has no post-G2 figures at 0.004% MMS because it gets blocked in G2 (e.g., in the dose-response experiment shown in Fig 2A: ~90% mononucleated dumbbell G2 cells; ~10% unbudded G1 cells; and <1% post-G2 cells). Because of that, it is impossible to make a chart of mph1Δ post-G2 figures at 0.004%. However, post-G2 cells in mph1Δ rose to 55% (±14%; SEM, n=3) at 0.001%, being this percentage equivalent to the % of post-G2 cells observed in the reference cdc15-2 and SSE mutants at 0.004% MMS. That is the reason we compared two different MMS concentrations (i.e., different concentrations but equivalent post-G2 EC50 among all these mutants).

Reviewer 2 Report

The manuscript written by Garcia-Luis was presenting FANCM/Mph1 helicase ortholog backup HR and stalled replication fork and blocking cells in G2 and not increasing anaphase bridge formation after MMS treatment,
This paper contains so many valuable data and well written,
But I have one trouble for a figure presentation with 2-y-axis graphs. It was not clear for me.

MMS concentration was described as v/v. Please use Molar.

Figure 2 does not have A in the figure.
It was not clear what is % cells of y-axis. Please clarify in the figure.
Then, what is % post-G2 cells of y-axis? 2 y-axis? It was not clear from legends. Please re-draw figures. This apply Figure 2A, 3A, 4A, 5A, 6A. 

Author Response

(authors’ replies in bold)

The manuscript written by Garcia-Luis was presenting FANCM/Mph1 helicase ortholog backup HR and stalled replication fork and blocking cells in G2 and not increasing anaphase bridge formation after MMS treatment. This paper contains so many valuable data and well written,
But I have one trouble for a figure presentation with 2-y-axis graphs. It was not clear for me.

1) MMS concentration was described as v/v. Please use Molar.

In the literature it is more often found “% v/v” as the way to represent the concentration of MMS, so we believe it is easier for readers to compare our results with others’ if we adhere to what it is standard in the field. Nevertheless, we now include the equivalence between “% v/v” and “M” in “Materials and Methods”.

2) Figure 2 does not have A in the figure.

Corrected.

3) It was not clear what is % cells of y-axis. Please clarify in the figure. Then, what is % post-G2 cells of y-axis? 2 y-axis? It was not clear from legends. Please re-draw figures. This apply Figure 2A, 3A, 4A, 5A, 6A. 

“% cells” refers to all cells counted, which are then categorized into G1, G2 or post-G2. The results of these quantifications are shown in the upper charts and, just the post-G2 percentage, in the grey 1st y-axis of the lower charts. “%post-G2 cells” refers to segregation categories within this subset of cells (whose percentage is set to 100% in the 2nd y-axis of the lower charts). In the lower charts, intended to show segregation defects of post-G2 cells under increasing concentrations of MMS, we still preferred to show in grey post-G2 cells vs all cells because it is easier to visualize that there is a direct correlation between incomplete segregation and proximity of the G2/post-G2 transition. We now explain this better in the figure legend where these charts first appear (Figure 2).